# The Pilot Study on Detecting Perforation with Abdominal Ultrasound During Gastric Endoscopic Submucosal Dissection

**DOI:** 10.3390/diagnostics15030335

**Published:** 2025-01-31

**Authors:** Ji Eun Kim, Jeayoun Kim, Tae Se Kim, Yang Won Min, Hyuk Lee, Byung-Hoon Min, Jun Haeng Lee, Poong-Lyul Rhee, Jae J. Kim

**Affiliations:** 1Department of Medicine, Samsung Medical Center, Sungkyunkwan University School of Medicine, Seoul 06351, Republic of Korea; taese.kim@samsung.com (T.S.K.); yangwon.min@samsung.com (Y.W.M.); lhyuk.lee@samsung.com (H.L.); jason.min@samsung.com (B.-H.M.); jh2145.lee@samsung.com (J.H.L.); pl.rhee@samsung.com (P.-L.R.); 2Department of Anesthesiology and Pain Medicine, Samsung Medical Center, Sungkyunkwan University School of Medicine, Seoul 06351, Republic of Korea; jeayoun.kim@samsung.com

**Keywords:** endoscopic submucosal dissection (ESD), abdomen ultrasound, gastric adenoma, early gastric cancer (EGC), perforation

## Abstract

**Objectives:** The indications for endoscopic submucosal dissection (ESD) for gastric adenoma and gastric cancer have expanded, leading to an increase in the number of patients with high procedural complexity. Post-ESD perforations prolong hospital stays and increase costs. However, no studies have focused on detecting micro-perforations during ESD. This study aimed to identify signs of perforation using abdominal ultrasound during gastric ESD. **Materials and methods:** This pilot study analyzed 50 patients who underwent abdominal ultrasound (VScan Air™, GE Healthcare) during ESD at Samsung Medical Center (March 2023–July 2024). Perforation was assessed via ultrasound, and post-procedure X-rays were performed for three days to detect free air. **Results:** Among 50 patients (median age 60, 76.1% male), the median procedure time was 60 min. Lesions were most common in the antrum (30.4%) and lesser curvature (17.4%). Pathology revealed 32.6% well-differentiated and 10.9% moderately differentiated adenomas, with 15.2% showing high-grade dysplasia. Free air was detected in three patients after procedures involving the body wall of the stomach. Abdominal US showed indirect signs of perforation, including an abnormal peritoneal line, hyperechoic shadowing, and an absence of normal gas patterns, confirmed by X-ray. **Conclusions:** Abdominal US is a simple, useful tool for rapid detection of perforation during ESD, enabling timely intervention. Further multicenter studies are needed to confirm these findings.

## 1. Introduction

Endoscopic submucosal dissection (ESD) is now a standard treatment for early gastric cancer (EGC) and gastric adenoma, often replacing surgery in selected cases [1,2,3,4,5]. Despite its efficacy, ESD carries risks of serious complications, such as perforation [6,7]. The incidence of intraoperative perforation ranges from 1.2% to 5.2% [7]. Small, localized gastric cancers without submucosal invasion are ideal for ESD, but lesions with uncertain features, such as slight submucosal invasion or irregular surface morphology, present challenges in regard to decision-making [8].

Predicting submucosal invasion prior to ESD has been a focus of many studies, but accurate detection remains complex [6,9]. Perforations during ESD are categorized as early or delayed. Early perforations are detected immediately using intraoperative imaging, while delayed perforations appear days later [10]. Factors such as tumor location, size, and technical aspects of the procedure have been associated with increased perforation risk [11,12]. Immediate identification and management of these perforations are crucial, as they can significantly impact patient outcomes, leading to prolonged hospitalization, delayed recovery, and increased healthcare costs [13].

Detecting perforations during ESD is challenging, partly due to the cautery effect. [11]. Excessive thermal injury during cauterization can obscure small perforations by creating a burn-like surface that is hard to distinguish from normal tissue or minor defects [14]. Additionally, prolonged procedures may lead to unnoticed air accumulation in the peritoneal cavity, complicating the detection of micro-perforations [15]. This highlights the need for additional tools, such as abdominal ultrasound (US), to detect subtle perforations in real time [16,17,18]. Abdominal US could offer a non-invasive and efficient tool for recognizing perforation during ESD, potentially enabling earlier intervention and reducing the risk of complications [19].

However, there has been no research on methods to try to detect micro-perforation during the procedure. This pilot study aims to evaluate the effectiveness of abdominal US for the real-time detection of perforation during gastric ESD with the goal of improving early intervention and patient prognoses.

## 2. Materials and Methods

This pilot study was conducted as a prospective cohort study at Samsung Medical Center between 1 March 2023 and 31 July 2024. Patients diagnosed with gastric adenoma or EGC who were scheduled to undergo an ESD performed by endoscopists with less than one year of experience were considered for inclusion. Initially, 89 patients were enrolled in the study. Abdominal US was performed by a well-trained expert. Patients who provided consent for study participation but for whom an ultrasound could not be physically performed during the procedure were excluded. Patients who were unable to undergo abdominal US were excluded, resulting in a final cohort of 50 patients. The study protocol was reviewed and approved by the Institutional Review Board of the Samsung Medical Center (IRB No. SMC 2023-01-029-002).

### 2.1. Inclusion and Exclusion Criteria

Patients who consented to undergo abdominal US during the ESD procedure were included. Patients who withdrew consent or did not undergo abdominal US during the procedure were excluded. Of the 89 initially enrolled patients, 6 withdrew consent and 33 did not undergo abdominal US, leading to 50 patients being included in the final analysis (Figure 1).

### 2.2. Procedure

During the ESD procedure, a portable abdominal US (VScan Air™, GE Healthcare, Chiacago, IL, USA) was utilized to detect signs suggestive of perforation. US findings, particularly the presence of free air, were documented in real time. Post-procedure, all patients underwent routine management, including X-ray imaging, for three consecutive days to confirm the presence of free air as an indicator of perforation. During the ESD procedure, if a clearly visible target sign was observed, the patient was planned to be excluded from the study and preventive clipping was to be performed.

### 2.3. Data Collection and Outcome Measures

The primary outcome of this study was the concordance between signs of perforation detected by abdominal US during the ESD procedure and the presence of free air during post-procedure X-ray imaging.

The secondary outcomes assessed in the study included lesion characteristics such as size, location, and histopathological features. Procedure-related factors, including procedure duration, were also analyzed. The incidence of perforation and its immediate management were documented. To evaluate diagnostic accuracy, histopathological findings were compared before and after the procedure. Collected variables included patient age, sex, procedure time, lesion location, endoscopic size, pathological size, and histopathological findings both pre- and post-procedure (Table 1).

### 2.4. Statistical Analysis

The values are expressed as the median (interquartile range) or mean for continuous variables and number (%) for categorical variables. All statistical analyses were performed using SPSS Statistics ver. 27.0 (IBM Corp, New York, NY, USA).

## 3. Results

A total of 50 patients were included in the final analysis, with a mean age of 65.7 ± 9.7 years, of whom 71.4% were male. The median procedure time was 60 min. Lesions were most commonly located in the antrum (44%), followed by the lower body (28%). The mean endoscopic size of the lesions was 21.3 ± 7.3 mm, while the mean pathological size was 10.7 ± 6.6 mm.

### Primary Outcome

The primary outcome assessed the concordance between perforation detected by abdominal US during ESD and post-procedure X-ray imaging. Abdominal ultrasound detected signs of perforation in three patients (6%) (Figure 2), which was confirmed by the presence of free air in post-procedural X-rays (Figure 3, Figure 4 and Figure 5). These findings included abnormal peritoneal strip, hyperechoic posterior shadowing, and the absence of normal gas patterns in the ultrasound (Figure 2). Figure 3A shows an ulcer following the ESD procedure. The lesion was located in the lower body anterior wall (LB AW), and no definite perforation was visible upon gross examination. Abdominal US confirmed the presence of a peritoneal strip, leading to clipping (imaging not shown). Figure 3B illustrates post-procedural free air. The lesion was confined to the lamina propria, and, histopathologically, it was identified as a poorly cohesive carcinoma, non-signet ring cell (SRC) type. Figure 4A also displays an ESD ulcer in a lesion located at the lower body, anterior wall, greater curvature. As shown in Figure 4B, severe fibrosis hindered the visualization and made dissection difficult. Due to the presence of fibrosis, the lifting layer was not well defined, making it difficult to establish a clear cutting line. Hyperechoic posterior shadowing was identified during the abdominal US, and, although no gross perforation was observed, clipping was performed at the deeply dissected area. Figure 4C presents a post-procedural free air X-ray. This patient’s lesion was also confined to the lamina propria, and the pathological result was tubular adenocarcinoma, moderate differentiated (MD) adenocarcinoma. Figure 5A shows an ESD ulcer in the posterior high body. Although there was a deep dissection, no clear target sign was observed. Severe fibrosis was present during the procedure, as seen in Figure 5B. An abdominal US performed during the procedure revealed a prominent peritoneal strip sign, without a normal bowel gas pattern, leading to clipping, as shown in Figure 5C. Figure 5D demonstrates a free air post-ESD. The lesion had invaded the submucosa by 300 µm, and an additional elective surgery was performed. A histopathological analysis revealed tubular adenocarcinoma, poorly differentiated (PD). Prompt intervention was undertaken in these patients, leading to successful management without severe complications.

### 3.1. Risk Analysis of Perforation

A detailed analysis of the patients who experienced perforation (*n* = 3) revealed common characteristics that may indicate potential risk factors. A histopathological analysis of the perforation cases showed a diverse range of lesion types, including poorly cohesive carcinoma, non-SRC, tubular adenocarcinoma, MD, and PD. The pathologic sizes of these lesions varied from 9 mm to 24 mm. While some US findings, like bright echogenicity and dirty shadowing, were not consistently observed, the commonality of the lesion location and histopathologic diversity highlight key areas for further investigation.

### 3.2. Lesion Characteristics and Other Outcomes

Overall lesion characteristics included a variety of histopathological findings. Before the procedure, the most common diagnoses included adenoma (28%), low-grade dysplasia (LGD, 14%), and high-grade dysplasia (HGD, 14%). Post-procedural pathology confirmed the initial diagnoses, with a majority being identified as well-differentiated adenocarcinoma (24%) and MD adenocarcinoma (34%).

## 4. Discussions

This pilot study explores a novel approach to the real-time detection of perforation during gastric ESD using abdominal US, emphasizing its potential to enhance procedural safety. Use of abdominal US enabled the detection of micro-perforations that could otherwise go unnoticed due to the visual limitations imposed by the cautery effect, a common challenge in ESD procedures. The cautery effect often obscures the field, making small perforations difficult to identify and increasing the risk of complications [20]. Conventional methods, such as post-procedural X-ray imaging, frequently result in delayed identification of perforations, thereby increasing the risk of complications like prolonged hospitalization, elevated healthcare costs, and worse patient outcomes. The use of real-time abdominal US, as demonstrated in this study, offers a promising alternative for earlier detection and timely intervention during ESD.

According to a comprehensive review evaluating complications of ESD in the upper gastrointestinal tract (GIT), studies analyzing factors influencing delayed perforation reported that no visible perforations were observed during the procedure. Cases requiring emergent surgery were predominantly due to delayed perforations that were not visibly detected at the time of the procedure [21]. Suzuki et al. reported that early detection of the onset of delayed perforations within 24 h after the procedure might be helpful in avoiding emergency surgery [10,12]. While it would be ideal to prevent delayed perforations, there is currently no definitive preventive method. Although complete closure of mucosal defects would theoretically be beneficial, it is often not feasible, particularly in certain lesions. Unlike the colon, the stomach has a thicker mucosal wall, and mucosal tissue does not always adhere completely during clip closure. Our study demonstrates that detecting perforations during the procedure using abdominal US can aid in preventing the need for emergency surgery. This approach provides a valuable opportunity for timely intervention, potentially improving patient outcomes.

In terms of risk factors for perforation, our study aligns with previous research that underscores the importance of tumor location. Lesions located in the LB AW were particularly prone to perforation, which is consistent with earlier findings about the technical challenges of performing ESD in certain gastric regions [22]. The LB AW presents unique anatomical complexities, such as a thinner walld and increased exposure to mechanical stress during dissection, making it more susceptible to perforation. These findings are in agreement with earlier studies, such as those by Roy Soetikno et al. (2005), who documented the difficulties associated with ESD complications in these anatomical areas [23].

The role of tumor size and depth of invasion also plays a critical part in predicting the risk of perforation. Our findings are consistent with the existing literature suggesting that larger tumors and those invading deeper into the submucosa are more prone to complications during ESD [23,24]. Given that these factors are often difficult to assess visually during the procedure, the use of abdominal US could serve as a valuable tool to enhance decision-making during the dissection process. By offering real-time, non-invasive monitoring, abdominal US may complement endoscopic findings, helping to mitigate the risk of complications such as perforation in technically challenging cases.

Advancements in diagnostic imaging tools, such as Magnetic Resonance Imaging and computed tomography protocols, have reduced interobserver variability, leading to a diminished role for abdominal US in recent years. However, abdominal US is now re-emerging as a valuable non-invasive tool, particularly in patients with inflammatory bowel disease. It can detect bowel wall thickening, peri intestinal inflammation, and extraluminal complications such as fistulas and abscesses [25,26]. Technological advancements, including high-frequency probes and color Doppler, have improved image quality and the ability to assess parietal vascularization [27]. It has gained significant attention recently as an imaging tool that can provide real-time visualization, it is not limited by location, and it is free from radiation exposure [28]. This study represents the first attempt to use abdominal US in real time during gastric ESD, offering a new approach to detecting intra-procedure complications like perforation. Unlike EUS, which is primarily a diagnostic tool used pre-procedure, abdominal US can provide continuous feedback during the procedure, enhancing the safety of the dissection. Abdominal US, as demonstrated in this study, can offer a more dynamic, intraoperative assessment of complications like perforations, especially in technically challenging procedures. This real-time utility provides a clear advantage over traditional preoperative imaging tools and could be particularly beneficial in cases where the tumor’s location or characteristics present higher risks in regard to perforation.

However, this study has several limitations. First, the relatively small sample size may limit the generalizability of the results. One of the reasons for the small sample size was that this study is a pilot study, and there have been no prior studies examining outcomes similar to ours. As a result, there were no predefined expectations for diagnostic rates or discrepancies. Second, as the procedures were performed by a single endoscopist with less than one year of experience, the outcomes may differ from those of more experienced practitioners. The level of endoscopist expertise is a known factor influencing complication rates during ESD, including perforation. Additionally, while this study focused on the utility of abdominal ultrasounds in detecting perforations, their potential role in preventing other complications, such as bleeding, requires further investigation. However, it is important to note that this was a pilot study, and even ESD experts are not exempt from complications. Despite the fact that both novices and experts experience complications, there has been little effort to detect and manage them in real time, which often leads to unnecessary healthcare costs. This study highlights a novel approach by emphasizing the potential of real-time detection and intervention, which has previously been overlooked.

To detect and prevent perforations in real time during procedures, future studies should involve large-scale prospective research. Such studies should be well designed, incorporating evaluations by multiple ultrasound experts and systematically observing procedures performed by endoscopists with varying levels of expertise.

In conclusion, this pilot study demonstrates the potential of abdominal US as a valuable tool for the early detection of perforations during ESD. By enabling timely intervention, the use of abdominal US can significantly improve patient outcomes and reduce the risk of serious complications such as delayed perforation or peritoneal seeding. Furthermore, the introduction of abdominal US into the standard ESD protocol could minimize the need for more invasive diagnostic tools post-procedure, thus optimizing the recovery process for patients. Future studies should aim to confirm these findings in larger cohorts and explore the broader applications of abdominal US in gastrointestinal endoscopy. Establishing clear guidelines for its integration into routine ESD procedures will be crucial for improving procedural safety and patient care.

## Figures and Tables

**Figure 1 diagnostics-15-00335-f001:**
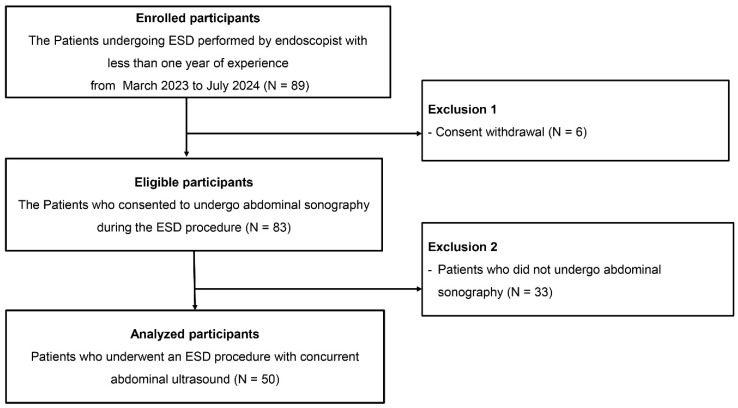
Flow chart of this study.

**Figure 2 diagnostics-15-00335-f002:**
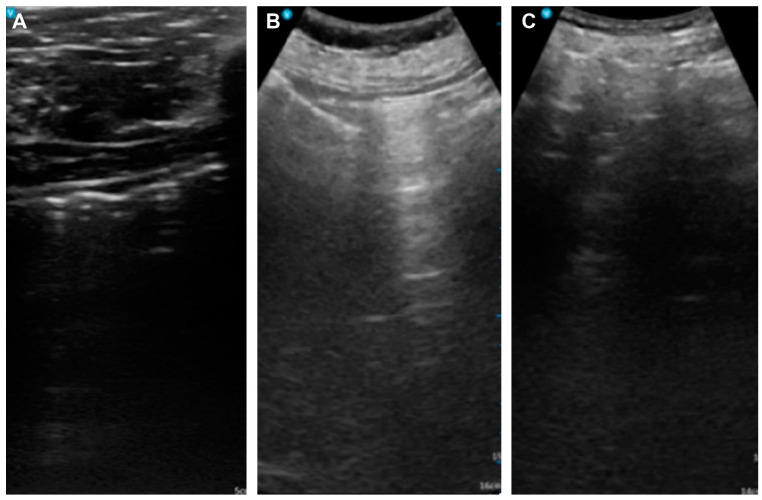
(**A**) Abnormal peritoneal strip. (**B**) Hyperechoic posterior shadowing not relevant with peristaltic movement. (**C**) Abnormal gas pattern.

**Figure 3 diagnostics-15-00335-f003:**
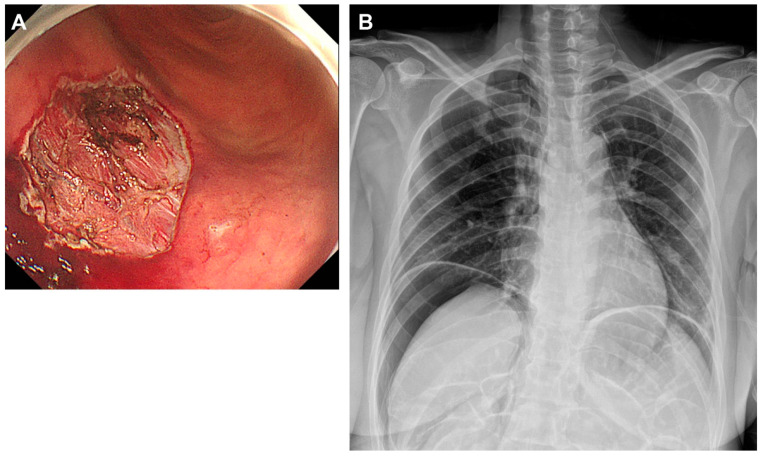
(**A**) ESD ulcer at the lower body, anterior wall (LB AW). (**B**) Post-procedure free air.

**Figure 4 diagnostics-15-00335-f004:**
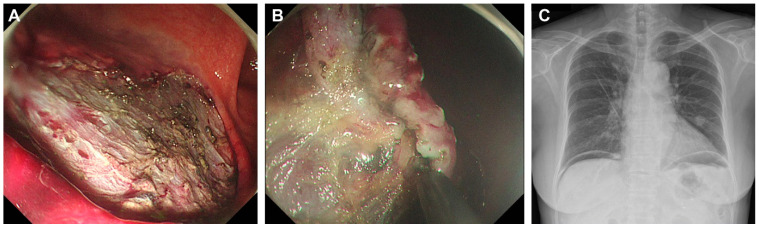
(**A**) ESD ulcer at the lower body, anterior wall (LB AW). (**B**) Severe fibrosis observed during ESD. (**C**) Post-procedure free air.

**Figure 5 diagnostics-15-00335-f005:**
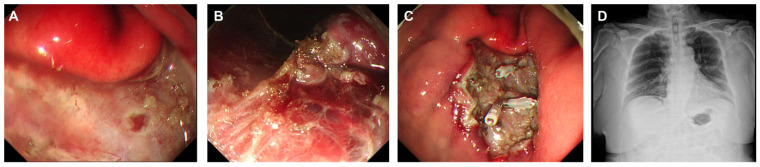
(**A**) ESD ulcer at the high body, posterior after clipping. (**B**) Fibrosis, (**C**) Clipping (**D**) Post-procedure free air.

**Table 1 diagnostics-15-00335-t001:** Baseline characteristics of enrolled patients.

Variable		Total (*n* = 50)
Age, mean ± SD		65.7 ± 9.7
Sex (male, (%))		35, 71.4%
Procedure time (min), median(IQR)		60 (40, 90)
Site of lesion (*n* =, %)	Antrum	22, 44.0
	LB	14, 28.0%
	MB	4, 8.0%
	HB	2, 4.0%
	Cardia	3, 6.0%
	angle	3, 6.0%
	Pylorus	2, 4.0%
Endoscopic size (mm), mean ± SD		21.3 ± 7.3
Pathologic size (mm), mean ± SD		10.7± 6.6
Forcep bx (*n*, %)	Adenoma, unclassified	7, 14.0%
	LGD	7, 14.0%
	HGD	7, 14.0%
	Adenocarcinoma, unclassified	2, 4.0%
	WD	16, 32.0%
	MD	5, 10.0%
	PD	0, 0%
	MD to PD	1, 2.0%
	Poorly cohesive carcinoma	2, 4.0%
	SRC	2, 4.0%
	NET	1, 2.0%
Post-procedure pathology (*n*, %)	LGD	5, 10.0%
	HGD	9, 18.0%
	Adenocarcinoma, unclassified	1, 2.0%
	WD	12, 24.0%
	MD	17, 34.0%
	PD	1, 2.0%
	SRC	1, 2.0%
	Poorly cohesive carcinoma	2, 4.0%
	NET	2, 4.0%
Perforation (*n*, %)		3, 6.0%

LB: lower body; MB: mid body; HB: high body; LGD: low-grade adenoma; HGD: high-grade adenoma; WD: well differentiated; MD: moderate differentiated; PD: poorly differentiated; SRC: signet ring cell carcinoma; NET: neuroendocrine tumor.

## Data Availability

The data underlying this article cannot be shared publicly, given the privacy expectations of the individuals who participated in the study. The data will be shared upon reasonable request to the corresponding author.

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
