# Peer review of "The Pilot Study on Detecting Perforation with Abdominal Ultrasound During Gastric Endoscopic Submucosal Dissection"

_diagnostics, 2025, doi:10.3390/diagnostics15030335_

Round 1

Reviewer 1 Report

Comments and Suggestions for Authors

Title: The Pilot Study on Detecting Perforation with Abdominal Ultrasound During Gastric Endoscopic Submucosal Dissection

Thank you for submitting your manuscript. After an in-depth review, I believe that considering the below major comments is necessary:

1.      Method part of the abstract should be summarized.

2.      You should add a comprehensive paragraph regarding “statistical analysis,” in method.

3.      Line 76: what does it mean by “Patients who 76 were unable to undergo abdominal US were excluded?” What was the reason?

4.      Line 119-121: Does any control diagnostic method (the gold standard) done for other patient? As, it is not a confirmed method, it is needed/ also it affects your following analysis.

5.      Line 156-161: “Risk analysis of perforation,” could not be done with this low number of positive cases; you could just suggest it for future studies.

6.      Line 174-178: the right place for conclusion is not here! Put it after discussion, please.

7.      Please revise the manuscript in term of being more clear. Do not use complex and uncommon words?

8.      Summarize the discussion please.

9.      Add a more comprehensive paragraph regarding “suggestions for future research” in discussion, as yours had several limitations!

Author Response

Comments from the Editors and Reviewers:

Reviewer #1

  1. Method part of the abstract should be summarized.

Answer : Thank you for your comments. We have revised the text as follows by removing redundant sentences. “Materials and methods: This pilot study is based on a prospective cohort registry of 89 patients diagnosed with gastric adenoma or early gastric cancer (EGC) at Samsung Medical Center between March 1, 2023, and July 31, 2024. Patients were recruited by an endoscopist with less than one year of ESD experience. After excluding 39 patients unable to undergo abdominal ultrasound (US), 50 patients who consented to US (VScan Air™, GE Healthcare, Illinois, USA) during ESD were ana-lyzed. Perforation was predicted using US, and all patients underwent X-ray imaging for three days post-procedure to check for free air.”

=>“ This pilot study analyzed 50 patients who underwent abdominal ultrasound (VScan Air™, GE Healthcare) during ESD at Samsung Medical Center (March 2023–July 2024). Perforation was assessed via ultrasound, and post-procedure X-rays were performed for three days to detect free air.”

  1. You should add a comprehensive paragraph regarding “statistical analysis,” in method.

Answer : The following has been added to the Methods section. “The values are expressed as the median (interquartile range) or mean for continuous variables and number (%) for categorical variables. All statistical analyses were performed using SPSS Statistics ver. 27·0 (IBM Corp, New York, NY, USA).”

  1. Line 76: what does it mean by “Patients who 76 were unable to undergo abdominal US were excluded?” What was the reason?

Answer : Thank you for your excellent comment. We apologize for not addressing this earlier. During the ESD procedure, the endoscopist must focus on the procedure itself, which necessitates that a specialized anesthesiologist proficient in ultrasound interpretation perform the ultrasound. While consent was obtained, there were instances where this was not feasible due to scheduling conflicts. We will add the following to the Methods section accordingly. “Abdominal US was performed by a well-trained expert. Patients who provided consent for study participation but for whom ultrasound could not be physically performed during the procedure were excluded.”

  1. Line 119-121: Does any control diagnostic method (the gold standard) done for other patient? As, it is not a confirmed method, it is needed/ also it affects your following analysis.

Answer : Thank you for your valuable comment. Performing an X-ray after the procedure is part of the routine check process, not only for all patients enrolled in this study but also for those who undergo procedures at our institution. However, in the case of abdominal ultrasound, its utility has not been previously evaluated, and despite being a specialized examination, there is no established reimbursement policy for it. What we aim to emphasize in this study is that it is a 'pilot' study, and we hope that these findings can serve as a basis for influencing future policy decisions.

  1. Line 156-161: “Risk analysis of perforation,” could not be done with this low number of positive cases; you could just suggest it for future studies.

Answer : I agree. To ensure the provision of objective information in the literature, the following content will be deleted as per your suggestion.

All patients with perforation were female, with ages ranging from 48 to 75 years. No-tably, the perforations were all associated with lesions located in the stomach's lower body area, an observation that aligns with the hypothesis that this region may present a higher risk of perforation during ESD.

And in discussion section, Page 8, lines 254-256.

“In conclusion, this pilot study demonstrates the potential of abdominal US as a valuable tool for the early detection of perforations during ESD., particularly in high-risk anatomical regions such as the lower body anterior wall.”

  1. Line 174-178: the right place for conclusion is not here! Put it after discussion, please.

Answer : We apologize for the oversight. The conclusions-related content, which was mistakenly included in the Results section, has been removed and appropriately addressed in the Discussion section.

“ 3.3. Conclusions

This study identifies potential risk factors for perforation during ESD, specifically in lesions located at the lower body anterior wall of the stomach. Abdominal US proved to be a valuable tool for the early detection and management of perforation, contributing to improved patient outcomes. Further multicenter or prospective studies are necessary to validate these findings.

  1. Please revise the manuscript in term of being more clear. Do not use complex and uncommon words?

Answer : Thank you for your feedback. We have reviewed and revised the complex and uncommon words as you suggested, and the edits have been directly reflected in the updated version of the manuscript, and have been highlighted.

  1. Summarize the discussion please.

Answer : Thank you for your comments. As you pointed out, we have revised and streamlined the lengthy content in the discussion section.

“This highlights the clinical significance of adjunctive tools like abdominal US for timely detection and management of perforation, especially in high-risk anatomical locations. The real-time feedback provided by abdominal US allows endoscopists to adjust their technique dynamically, potentially preventing full-thickness perforations before they progress to more severe complications.

Endoscopic Ultrasonography (EUS) has traditionally played an essential role in evaluating tumor invasion depth and determining the indications for ESD. Studies like those by Ye Han et al. (2016) have emphasized the efficacy of ESD in treating early gastric cancer (EGC) without lymph node metastasis [1], while Kuroki et al. (2020) demonstrated the high sensitivity and specificity of EUS in predicting the T stage of gastric cancer, achieving a 95% accuracy rate in differentiating between mucosal/superficial submucosal and deep submucosal invasion [2]. However, EUS has limitations, particularly in its ability to clearly distinguish between mucosal and submucosal cancers. As noted by Park and Lee (2016), this distinction remains controversial and subject to variability [3, 4]” Thus, while EUS plays a pivotal role in assessing lymph node involvement and invasion depth, its utility in real-time detection of complications during ESD is limited.

  1. Add a more comprehensive paragraph regarding “suggestions for future research” in discussion, as yours had several limitations!

Answer : Thank you very much for your valuable feedback to improve the manuscript. We have removed the future research section mentioned below and rephrased it into a comprehensive paragraph as suggested.

. Future studies should involve larger, multicenter trials with more experienced en-doscopists to validate the efficacy of abdominal US as a routine adjunct to ESD. Additionally, while this study focused on the utility of abdominal ultrasound in detecting perforations, its potential role in preventing other complications, such as bleeding, re-quires further investigation. However, it is important to note that this was a pilot study, and even ESD experts are not exempt from complications. Despite the fact that both novices and experts experience complications, there has been little effort to detect and manage them in real time, which often leads to unnecessary healthcare costs. This study highlights a novel approach by emphasizing the potential of real-time detection and intervention, which has previously been overlooked.

To detect and prevent perforations in real time during procedures, future studies should involve large-scale prospective research. Such studies should be well-designed, incorporating evaluations by multiple ultrasound experts and systematically observing procedures performed by endoscopists with varying levels of expertise.

Reviewer 2 Report

Comments and Suggestions for Authors

The paper aims to identify signs of perforation using abdominal ultrasound during gastric ESD.

My comments:

1. Please, in the abstract section add the relationship between ESD and cancer as the best justification for the study

2. In the introduction, please include the prevalence of perforation during ESD. Also, is it common in gastrointestinal surgeries?

3. Please, better explain why did you choose "endoscopists with less than one year of experience" in line 75.

4. The size sample was not calculated? Or it was a convenience sample?

5. Does severe fibrosis may impact the perforation visualization during ESD? I suggest a better explanation in the text.

6. Line 173. The conclusion section should be after the discussion section since the discussion helps to better conclude the results obtained in this study.

7. In the discussion, please insert information about the US and ESD guidelines to justify the importance of avoiding perforations.

8. Finally, better explain what are the clinical benefits of using US when compared to ESD?

Author Response

Reviewer #2:

  1. Please, in the abstract section add the relationship between ESD and cancer as the best justification for the study

Answer : Thank you for your valuable comments. We have revised the following to the abstract. (page1, lines 19-20)

 “Perforation during endoscopic submucosal dissection (ESD) is a serious complication requiring prompt diagnosis and management. The indications for endoscopic submucosal dissection (ESD) for gastric adenoma and gastric cancer have expanded, leading to an increase in the number of patients with high procedural complexity.”

  1. In the introduction, please include the prevalence of perforation during ESD. Also, is it common in gastrointestinal surgeries?

Answer : The J-WEB/EGC cohort revealed that intraoperative perforations during gastric ESD occurred in 2.3% of lesions (218/10,821); however, most of them could be treated with conservative management, and only seven cases required emergency surgery. We have added the following to the introduction[5] (page1, line 43).

 “The incidence of intraoperative perforation ranges from 1.2% to 5.2%.”

  1. Please, better explain why did you choose "endoscopists with less than one year of experience" in line 75.

Answer : Although experienced endoscopists are more skilled in procedures and better equipped to manage complications, there have been no studies providing clear data on the differences in perforation rates based on the endoscopic proficiency of experts during ESD. The reason for planning this pilot study on perforation was that, despite the severity of perforation as a complication, there has been little effort to address it. We limited the study to endoscopists with less than one year of experience to exclude outcomes influenced by knowledge and techniques acquired through diverse experiences, which are not scientifically established. Please refer to the following section on limitations for further details.

 “Despite the fact that both novices and experts experience complications, there has been little effort to detect and manage them in real time, which often leads to unnecessary healthcare costs.”

If this study demonstrates the utility of ultrasound during the procedure, it would be worthwhile to plan a prospective study in the future to investigate differences in incidence rates based on proficiency and the role of ultrasound in detecting such differences.

  1. The size sample was not calculated? Or it was a convenience sample?

Answer : Thank you for your excellent comment. We consulted with the statistics team, and since this is a pilot study, it was not deemed necessary to calculate precise statistical values for diagnostic rates, expectations, or discrepancies. However, your point regarding the use of a convenience sample is undeniable. We will include this in the limitations section. Please refer to the description provided in the manuscript (page 8, lines 235-238)

“ One of the reasons for the small sample size was that this study is a pilot study, and there have been no prior studies examining outcomes similar to ours. As a result, there were no predefined expectations for diagnostic rates or discrepancies.”

  1. Does severe fibrosis may impact the perforation visualization during ESD? I suggest a better explanation in the text.

Answer : Thank you for your detailed questions regarding the manuscript. As shown in the image below, when there is no fibrosis during the ESD procedure, the SM (submucosal) layer lifts well, allowing for better visualization and enabling the endoscopist to clearly establish the cutting line. However, as seen in Figure 4(B), if there is no lifting of the blue SM layer and white to yellowish fibrotic tissue is present, the absence of a lifting layer makes it difficult to distinguish between the mucosa and muscle layers, hindering precise cutting line determination. In Korea, this distinction is also reflected in reimbursement policies; if fibrosis is present, both the dual knife and needle knife are reimbursable under the established system. We have added the following content to the manuscript. (page5, line 134-135)

 “Due to the presence of fibrosis, the lifting layer was not well-defined, making it difficult to establish a clear cutting line.”

Gastrointest Endosc Clin N Am. 2023 Jan;33(1):67-81

  1. Line 173. The conclusion section should be after the discussion section since the discussion helps to better conclude the results obtained in this study.

Answer : We apologize for the oversight. The conclusions-related content, which was mistakenly included in the Results section, has been removed and appropriately addressed in the Discussion section.

“ 3.3. Conclusions

This study identifies potential risk factors for perforation during ESD, specifically in lesions located at the lower body anterior wall of the stomach. Abdominal US proved to be a valuable tool for the early detection and management of perforation, contributing to improved patient outcomes. Further multicenter or prospective studies are necessary to validate these findings.

  1. In the discussion, please insert information about the US and ESD guidelines to justify the importance of avoiding perforations.

Answer : The guidelines for gastric ESD provide recommendations regarding indications and techniques but do not address methods for managing complications. Effective management of complications is essential for improving patient outcomes, and as you mentioned, such guidance should be included in future guidelines. To address this gap, we have reviewed various studies on managing complications and have added the following content to the discussion section. (page 7, line 185-198)

 “According to a comprehensive review evaluating complications of ESD in the upper gastrointestinal tract (GIT), studies analyzing factors influencing delayed perforation reported that no visible perforations were observed during the procedure. Cases requiring emergent surgery were predominantly due to delayed perforations that were not visibly detected at the time of the procedure [6]. Suzuki et al. reported that early detection of the onset of delayed perforations within 24 hours after the procedure might be helpful in avoiding emergency surgery[7, 8]. While it would be ideal to prevent delayed perforations, there is currently no definitive preventive method. Although complete closure of mucosal defects would theoretically be beneficial, it is often not feasible, particularly in certain lesions. Unlike the colon, the stomach has a thicker mucosal wall, and mucosal tissue does not always adhere completely during clip closure. Our study demonstrates that detecting perforations during the procedure using abdominal ultrasound can aid in preventing the need for emergency surgery. This approach provides a valuable opportunity for timely intervention, potentially improving patient outcomes.”

  1. Finally, better explain what are the clinical benefits of using US when compared to ESD?

Answer : The indications for gastric ESD are gradually expanding. In the past, lesions with certain levels of differentiation or size were typically treated with gastric resection like surgery. However, these lesions are now often managed endoscopically. As a result, the complexity of the procedure has increased. Endoscopists must now pay even greater attention to invisible cutting lines, especially in cases involving fibrosis or submucosal (SM) invasion. During the procedure, resecting without a clearly defined cutting line can lead to unintended outcomes, such as deeper-than-expected resection depth or extensive cauterization, which might result in perforations that are not visually detectable. In such situations, the use of ultrasound becomes crucial. If ultrasound identifies findings suggestive of perforation, the procedure can be promptly concluded using a snaring technique, while thoroughly examining the suspected area for potential damage. Identified defects can then be addressed with clips or other closure methods. If a perforation remains undetected and the procedure continues as planned, intra-abdominal air accumulation can occur, leading to post-procedural complications such as abdominal pain or hemodynamic instability. Therefore, incorporating abdominal ultrasound during ESD allows for early detection of perforations, enabling timely emergency interventions.

Round 2

Reviewer 1 Report

Comments and Suggestions for Authors

It is accepted.